# d-Serine Increases Release of Acetylcholine in Rat Submandibular Glands

**DOI:** 10.3390/biology12091227

**Published:** 2023-09-11

**Authors:** Masanobu Yoshikawa, Migiwa Okubo, Kosuke Shirose, Takugi Kan, Mitsuru Kawaguchi

**Affiliations:** 1Department of Clinical Pharmacology, School of Medicine, Tokai University, Isehara 259-1193, Japan; 2Kawano Dental Clinic, Yachimata 289-1101, Japan; mohkub@sd6.so-net.ne.jp; 3Department of Anesthesiology, School of Medicine, Tokai University, Isehara 259-1193, Japan; sk327798@tsc.u-tokai.ac.jp (K.S.); tkan@is.icc.u-tokai.ac.jp (T.K.); 4Tokyo Dental College, Misaki-cho, Chiyoda-ku, Tokyo 101-0061, Japan; mtskawaguti@gmail.com

**Keywords:** d-serine, submandibular gland, acetylcholine release, saliva, in vivo microdialysis

## Abstract

**Simple Summary:**

Oral administration of d-serine, but not l-serine, increased its concentrations in the rat submandibular gland and pilocarpine-induced salivary secretion. Perfusion of the rat submandibular gland with d-serine via the submandibular gland artery increased carbachol-induced salivary secretion. The proportion of the d-form of serine in interstitial fluid was higher than that in plasma or saliva. Infusion of d-serine with l-glutamate through the microdialysis probe significantly increased acetylcholine levels in the submandibular glands in an *N-*methyl-d-aspartate receptor glycine site antagonist-sensitive manner.

**Abstract:**

d-serine has been observed in submandibular gland tissue in rats, but its functions remain to be clarified. Oral administration of d-serine, but not l-serine, increased its concentrations in the submandibular gland and pilocarpine-induced salivary secretion. In vivo microdialysis was used to collect the d- and l-enantiomers of amino acids from local interstitial fluid in the rat submandibular gland. The proportion of the d-form of serine in interstitial fluid was higher than that in plasma or saliva. Perfusion of the rat submandibular gland with d-serine and l-glutamic acid via the submandibular gland artery resulted in a significant increase in salivary secretion after stimulation of muscarinic receptors with carbachol. In vivo microdialysis applied to the submandibular glands of rats showed that infusion of d-serine along with l-glutamate through the microdialysis probe significantly elevated acetylcholine levels in local interstitial fluids in the submandibular glands of anesthetized rats as compared to that with l-glutamate alone in an *N*-methyl-d-aspartate receptor glycine site antagonist-sensitive manner. These results indicate that d-serine augments salivary secretion by increasing acetylcholine release in the salivary glands.

## 1. Introduction

A growing body of evidence has accumulated showing that endogenous d-serine is an obligatory co-agonist for the glycine site of the *N*-methyl-d-aspartate receptor (NMDA receptor) in the mammalian brain [1,2,3]. In the forebrain, where NMDA receptors are abundant, considerable amounts of d-serine were detected in extracellular fluids, indicating that d-serine is involved in glutamatergic neurotransmission via these receptors. Regional profiles of serine racemase, which converts l-serine to d-serine, are similar to those of endogenous d-serine and NMDA receptors, with higher levels in the forebrain and lower levels in the hindbrain [4,5]. On the other hand, d-amino acid oxidase (DAO, EC 1.4.3.3), which catabolizes d-serine to alpha-keto acids, ammonium ions, and hydrogen peroxide, is located primarily in the hindbrain [6,7,8].

Compared to the forebrain, d-serine has been found in considerably lower amounts in most peripheral tissues after birth. In the rat submandibular gland, d-serine (4.9 nmol/g wet tissue) [9] amounted to approximately 2% of that observed in the cerebral cortex [10]. Protein expressions of serine racemase and DAO in rat submandibular glands amounted to approximately 20% and 15%, respectively, of those observed in the cerebral cortex [9]. Protein expression of the NR1 subunit, which is essential for ion channel assembly and NMDA receptor functional activity, amounted to approximately 1% of that observed in the cerebral cortex [9]. Triheteromeric NR1/2A/2B receptors are the most abundant NMDA receptors in the forebrain [11], whereas only diheteromeric NR1/2D receptors were exclusively detected in the three major salivary glands of the rat [9]. A type of triheteromeric NMDA receptor, which contains two NR1 and two distinct NR2 subunits, has been reported to mediate NMDA-induced toxicity [12]. Rod bipolar cells, which express diheteromeric NR1/2D receptors as in salivary glands, are resistant to excitotoxicity by calcium influx [13,14]. These findings correspond well with previous results demonstrating higher concentrations of normal extracellular Ca^2+^ in salivary glands (2.56 mM) [15] than in the hippocampus (0.5 mM) [16]. Taken together, the fact that different expression of NR2 subunits in the submandibular gland and in the forebrain is characteristic of salivary gland function may support the notion that different NR2 subtypes mediate different physiological functions [17,18].

In salivary glands, parasympathetic and sympathetic nerves are responsible for salivary water secretion and protein secretion, respectively. Acetylcholine released from the parasympathetic nerve terminals in the salivary glands binds to primarily M3 muscarinic receptors on the acinar cells, leading to the release of calcium stored in the endoplasmic reticulum. Elevated intracellular calcium activity opens chloride and potassium channels in the acinar cells, resulting in the secretion of plasma water as saliva [19,20].

To clarify the roles of d-serine in the control of salivary gland functions, we have presently investigated the effects of d-serine on saliva secretion and on acetylcholine release in submandibular glands.

## 2. Materials and Methods

### 2.1. Animals

Male Wistar rats (8–9 weeks old, 230–280 g body weight, *n* = 176; Nihon Clare, Tokyo, Japan) were bred in an air-conditioned room with a temperature of 24–26 °C and humidity of 50–60% on a 12-h light/dark cycle (lights on at 07:00 h, food and water available ad libitum). Animals were allowed to adapt to the new experimental environment for one week.

### 2.2. Chemicals

The amino acids were purchased from Sigma (St. Louis, MO, USA), Tokyo Kasei (Tokyo, Japan), Nacalai Tesque (Kyoto, Japan), and FUJIFILM Wako Chemical Co. (Osaka, Japan). The following were obtained from the sources indicated: o-phthaldialdehyde (OPA; Nacalai Tesque, Kyoto, Japan), *N*-tert-butyloxycarbonyl-L-cysteine (Boc-l-Cys; Sigma, St. Louis, MO, USA), and d-amino acid oxidase (DAO; Sigma, St. Louis, MO, USA). Unless otherwise indicated, all chemicals were purchased from FUJIFILM Wako Chemical Co. (Osaka, Japan).

### 2.3. Sample Preparation for the Tissues 

Rats were euthanized by exsanguination from the abdominal aorta under anesthesia with pentobarbital (50 mg/kg, intraperitoneal administration). The submandibular glands were quickly excised and stored at −80 °C until use. Free amino acids were extracted using a method similar to the one described previously [21].

### 2.4. Sample Preparation for Plasma and Saliva

After cardiac blood collection, blood was collected in ethylenediaminetetraacetic acid-containing tube (BD Microtainer^®^ MAP; Becton-Dickinson, Franklin Lakes, NJ, USA) under anesthesia with pentobarbital (50 mg/kg, intraperitoneal administration). The samples were centrifuged at 3500× *g* for 10 min at room temperature. The obtained supernatant was used as plasma sample. Apart from these plasma collection experiments, the oral cavities of rats were cleaned by small cotton balls soaked in saline under anesthesia with pentobarbital (50 mg/kg, intraperitoneal administration). The rats were then injected with pilocarpine (1.0 mg/kg, subcutaneous administration) to stimulate salivary secretion. The saliva was subsequently collected from the oral cavity using a micropipette and placed in tubes in an ice bath. These samples of plasma and saliva were stored at −80 °C until use. Equal volume of 10% trichloroacetic acid (TCA) was mixed with the samples of plasma or saliva, followed by centrifuging at 10,000× *g* for 10 min at 4 °C. To remove TCA, the supernatant was washed with water-saturated diethyl ether. Thirty µL of the aqueous layer was dried under reduced pressure with a rotary evaporator (CVE-3100; Eyela, Tokyo, Japan) at 40 °C.

### 2.5. Sample Preparation of Interstitial Fluids of Submandibular Glands

The rats were anesthetized with isoflurane (2%) by inhalation. The interstitial fluids of the submandibular gland of rats were collected by in vivo microdialysis using a method similar to the one described previously [22]. Briefly, during in vivo microdialysis experiments, the rectal temperature of rats was maintained at approximately 38 °C by a body temperature maintenance device (Muromachi Kikai Co., Tokyo, Japan). Continuous intraperitoneal infusion (0.5 mL/h) of saline solution was performed by a pressurized pharmaceutical infuser (FC-PCA III, AuBEX, Tokyo, Japan) (Figure 1). The concentrations of the amino acids in the dialysate were measured by HPLC (high performance liquid chromatography). Raising the perfusion speed from 1 µL/min to 5 µL/min decreased the relative recovery of any of the amino acids targeted [(concentration in dialysate)/(concentration in testing solution)]. In contrast, the absolute recovery rate [(concentration in dialysate) × (perfusion speed)] varied nonlinearly, reaching a maximum at 2 µL/min (Figure 2A–C). Using this perfusion speed (2 µL/min) in vitro (Figure 2D–F), a nearly uniform relative recovery rate was obtained, even when the amino acid concentration in the testing solution was changed. Based on these results, the Ringer’s solution was perfused into the probe at a rate of 2 μL/min by a micro-syringe pump (ESP-32, Eicom, Kyoto, Japan). One sampling period was 15 min, which allowed for collecting sufficient amino acids for satisfactory quantitative determination. Each 15 min effluent was collected in chilled polypropylene tube on a micro-fraction collector (EFC-96, Eicom, Kyoto, Japan). Figure 3 shows the time course of amino acid levels in the dialysate collected at 15 min intervals over a period of 240 min. After implantation of the probe, the amino acid levels were variable and unstable over the first 120 min (Figure 3). This activity gradually decreased, however, and levels stabilized thereafter. Dialysates collected between 120 and 240 min after probe insertion were used for amino acid analysis of interstitial fluid. These dialysates were stored at −80 °C until use. For amino acid analysis, the dialysates were concentrated with a vacuum and centrifugal dehydrator evaporated (CVE-3100; Eyela, Tokyo, Japan) at 40 °C.

### 2.6. Derivatization of Amino Acids with Boc-l-Cys-OPA 

Derivatization reagent (Boc-l-Cys-OPA) was prepared daily by dissolving 20 mg of OPA and 20 mg of Boc-l-Cys in 2 mL of MeOH. Each sample of plasma, saliva, and dialysate was dissolved in 400 mM sodium borate buffer (pH 9.0). To 15 µL of the sample solution, 10 µL of Boc-l-Cys-OPA reagent was added. After derivatization for 2 min at room temperature, 20 µL of the reaction mixture was introduced onto the HPLC system. 

### 2.7. HPLC System for Determination of d- and l-Enantiomers in the Plasma, Saliva, and Dialysate

The simultaneous determination of d- and l-enantiomers of serine and glutamate in samples was accomplished using high-performance liquid chromatography (HPLC) as previously described [21]. Briefly, the HPLC system consisted of DGU-20A3R degasser (Shimadzu, Kyoto, Japan), LC-20AD pump (Shimadzu, Kyoto, Japan), CTO-20AC column oven (Shimadzu, Kyoto, Japan), FP-4020 fluorescence detector (Jasco, Tokyo, Japan), and M510S auto-sampling injector (Eicom, Kyoto, Japan). d-Serine was identified based on retention time and a well-resolved peak and further confirmed by loss of the peak due to degradation by addition of DAO (2–5 units for tissue, plasma, saliva; 1 unit for dialysate).

### 2.8. In Situ Perfusion of Rat Submandibular Glands

In situ perfusion systems that maintain physiological functions such as neuronal control were accomplished as previously described [23].

### 2.9. Collection and Determination of Acethylcholine in Interstitial Fluids of Submandibular Glands

The rats were anesthetized with isoflurane (2%) by inhalation. The collection and determination of acethylcholine in interstitial fluids of rat submandibular glands was accomplished using in vivo microdialysis and HPLC as previously described [22].

### 2.10. Statistical Analyses

The results are presented as the mean and standard deviation (SD). A statistical analysis software package (GraphPad Prism, version 9.0, GraphPad Software, San Diego, CA, USA) was used for comparison across experimental conditions. Dunn’s multiple comparison post hoc tests were used to determine significance in each group when a significant difference among groups was obtained by Kruskal–Wallis tests or Friedman tests for more than two groups. Dunn’s comparison post hoc tests were used to identify differences among groups when a significant difference in *F* value was observed in the two-way analysis of variance (ANOVA). A *p*-value less than 0.05 was considered to indicate statistical significance.

## 3. Results

### 3.1. Time Course of the Changes in d- or l-Serine Level in Plasma and Submandibular Glands after Oral Administration

The concentrations of d-serine in plasma and submandibular gland increased rapidly from 15 to 30 min after oral administration of d-serine (1 mmol/kg) and decreased gradually after 60 min. The concentrations of d-serine levels in the submandibular gland showed changes over time that were approximately twofold greater than those in plasma. (Figure 4A). On the other hand, the concentrations of l-serine in both plasma and the submandibular gland remained stable after l-serine administration (1 mmol/kg) (Figure 4B).

### 3.2. Effects of Oral Administration of d- or l-Serine on Salivary Secretion under Pilocarpine Stimulation

Oral administration of d-serine (100–2000 µmol/kg) significantly increased pilocarpine-induced salivary secretion in a dose-dependent manner, whereas l-serine (100–2000 µmol/kg) did not change it (Figure 5).

### 3.3. d- and l-Serine in Plasma, Saliva, and Interstitial Fluid

The concentration of d-serine was determined in plasma, saliva, and interstitial fluid in the submandibular gland. Small amounts of d-serine were detected in saliva (0.7 µM) and interstitial fluids (0.2 µM) compared to plasma (2.5 μM) (Table 1). Assuming that the in vivo recovery rate is comparable with the in vitro recovery rate (d-serine: 49.4%; l-serine: 49.9%; l-glutamate: 47.8%), it may be possible to estimate the extracellular levels of amino acids in rat submandibular glands using the latter. The results for interstitial fluid in Table 1 represent the fraction of dialysates collected between 120 and 180 min after probe insertion. The proportion of the d-form of serine in the interstitial fluids (3.6%) was high in comparison with plasma (1.2%) or saliva (1.1%).

### 3.4. Effects of Perfusion of d-Serine into Submandibular Gland on Carbachol-Induced Salivary Secretion

Carbachol (10 µM) was applied every 30 min for 5 min. Prior to carbachol administration, l-glutamate with or without the indicated concentration of d-serine was administered for 5 min, and saliva from the submandibular gland was collected every 5 min (Figure 6A). Based on the results of amino acid analysis (Table 1), the concentrations of l-glutamate (150 µM) and d-serine (2 µM) in perfusates were defined as physiological ones, respectively. After muscarinic receptor stimulation with carbachol, perfusion of the rat submandibular gland with d-serine (2–200 µM) resulted in an increase in salivary secretion in a dose-dependent manner up to 100-fold higher than physiological concentrations (Figure 6B). In contrast, perfusion of the submandibular gland with a high dose of l-glutamate (15 mM) alone, which is 100-fold higher than the physiological concentration, drastically reduced salivation (Figure 6C).

### 3.5. Effects of In Vivo Infusion of d-Serine into Submandibular Gland on Acetylcholine Levels

Figure 6 shows changes in acetylcholine levels in the dialysate due to d-serine in the presence of l-glutamate. Based on the results of amino acid analysis (Table 1) and recovery rate (d-serine: 49.4%, l-glutamate: 47.8%) (Figure 2), the concentration of l-glutamate (40 µM) and d-serine (0.5 µM) in perfusates was defined as the physiological concentration of interstitial fluids, respectively. Infusion with 10 µM or 20 µM d-serine yielded a significant increase in the extracellular levels of acetylcholine by approximately twofold of the basal level (*p* < 0.05) (Figure 7A). Upon termination of d-serine perfusion, the levels returned to basal levels. Infusion of R-(+)-HA-966, a water-soluble antagonist/partial agonist at the glycine site of the NMDA receptor, suppressed the effects of d-serine on the increase in extracellular levels of acetylcholine (Figure 7B).

## 4. Discussion

The present study shows that d-serine and l-glutamate act on the rat submandibular gland, resulting in increased saliva after stimulation of muscarinic receptors with carbachol. This is not consistent with the results of a previous study that demonstrated that infusion of l-glutamate (1–100 mM) or NMDA (1–100 mM) into the rat submandibular gland had no effect on saliva secretion induced by electrical stimulation of the chorda-lingual nerve [24]. This discrepancy could be due to the following reasons: First, stimulation of the glycine binding site on the NMDA receptor is essential for adequate neurotransmission to be generated by glutamate. Thus, glycine site agonists such as d-serine are designated as co-agonists of the NMDA receptor [25,26]. In fact, the regional profiles of serine racemase in the brain are similar to those of endogenous d-serine and NMDA receptors [4,27,28]. Second, the glycine site of the NMDA receptor is not saturated under physiological conditions, even though d-serine exceeds the extracellular concentration that fully activates the glycine portion of the NMDA receptor [29]. This is in good agreement with the results in the present study that oral administration of d-serine (1 mmol/kg) reached approximately 600 µM in plasma, approximately 300 times more than the physiological condition, resulting in a significant increase in salivary secretion (Table 1, Figure 4 and Figure 5). Third, glutamatergic neurotransmission via the NMDA receptor is required to maintain the extracellular concentration of glutamate within an adequate range [30,31,32,33]. High concentrations of glutamate result in neurotoxicity. This notion supports the result in this study showing that a high dose of l-glutamate (15 mM) decreased salivary secretion after stimulation by carbachol. In rat submandibular glands, the extracellular contents of glutamate (approximately 16 µM; Table 1) may be saturated for activating the glutamate site of the NMDA receptor.

The present study demonstrated that perfusion and infusion of d-serine into submandibular glands increased salivary secretion and released acetylcholine levels, respectively. This is in good agreement with previous studies on peripheral organs such as the cochlea: (1) in the cochlea, the effects of d-serine on auditory function are physiological as a result of its modulation of synaptic transmission; (2) d-serine acts specifically on the postsynaptic auditory neurons without altering the functions of the cochlea; and (3) d-serine produced robust calcium responses induced by activation of the NMDA receptor in spiral ganglion neurons in the cochlea. These results indicated that d-serine plays a role in promoting activation of the NMDA receptor in the cochlea in physiological conditions [34]. Furthermore, the present results are in good agreement with previous studies showing that the NMDA receptors are located presynaptically and that acetylcholine is released by NMDA receptor stimulation in the brain [35,36,37,38]. Further studies on the localization of NMDA receptors in the salivary glands are needed.

Initially considered to be exclusively expressed in astrocytes in the brain [4], serine racemase was later found to be primarily expressed in neurons under physiological conditions, in particular in glutamatergic and GABAergic neurons [39,40,41]. Pathological conditions shift d-serine production from neurons to primarily glial cells. Inflammatory stimuli, including amyloid β-peptide and lipopolysaccharide, increased transcription levels of serine racemase through two AP-1 elements, leading to neuronal damage through excessive activation of NMDA receptors [42,43]. In peripheral tissue such as the cochlea, it has been demonstrated that neonatal serine racemase knockout mice did not affect hearing function but were resistant against noise-induced permanent hearing loss, suggesting that d-serine plays a role in promoting activation of the NMDA receptor in pathological conditions [34].

The important question arises as to whether cells release endogenous d-serine. Previous studies demonstrated that serine racemase and d-serine are located in neurons as well as astrocytes [28,44,45]. Under physiological conditions, d-serine acts as a neurotransmitter and is involved in the mediation of synaptic plasticity by NMDA receptors. In contrast, under pathological conditions such as injury, d-serine release is switched from neurons to astrocytes [46]. Previous immunohistochemistry studies showed expression of the nicotinic α7 receptor on myoepithelial cells that surround the acini and intercalated ducts in the submandibular glands of the non-obese diabetic mouse model developing Sjögren’s syndrome features, such as lymphocytic infiltrates in the salivary glands and reduction of salivary gland function [47]. Parasympathetic stimulation of the salivary glands generally leads to increased saliva excretion but also stimulates contraction of the myoepithelial cells to enhance saliva expulsion [48]. Interestingly, activation of the nicotinic α7 receptor potentiates d-serine synthesis and/or release from astrocytes, resulting in the potentiation of the functions of NMDA receptors in the brain [49,50,51,52,53,54]. Further studies may reveal the interaction between d-serine release from myoepithelial cells and salivary gland dysfunction associated with inflammation. 

Modulation of cholinergic neurons mediated by glutamate receptors in the brain has been extensively studied in the context of learning, memory, and neurodegeneration. The activities of the cholinergic neurons are differentially regulated by the type of glutamate receptor involved and the brain region [55,56,57,58,59,60,61]. Previous studies demonstrated that activation of the glutamate site [55], the Mg^2+^ binding site [57,59], or the glycine site [62] of NMDA receptors increases acetylcholine release in the striatum. The present study showed that d-serine increased extracellular levels of acetylcholine in submandibular glands through the glycine site of the NMDA receptor. To the best of our knowledge, this is the first study to detect an increase in acetylcholine release by activation of the NMDA receptor in peripheral tissue. 

In contrast to the NMDA receptor, activation of the a-amino-3-hydroxy-5-methyl-4-isoxazolepropionic acid (AMPA) receptor decreased acetylcholine release via the gamma-aminobutyric acid (GABA) receptor in the brain [57,58,59]. This is consistent with the results of studies on salivary glands, as follows: First, infusion of AMPA (100 mM) into the rat submandibular gland inhibited saliva secretion induced by electrical stimulation of the chorda-lingual nerve [24]. Second, GABA and its biosynthetic and metabolic enzymes exist in rat salivary glands [63,64]. Third, stimulation of GABA receptors on acinar cells of rat sublingual glands causes chloride ions to flow into the cells [65]. Fourth, infusion of GABA via the submandibular gland artery decreased salivary secretion [23]. These suggest that the regulation of cholinergic neurons via the glutamatergic system in the brain has a similar underlying mechanism in the salivary glands.

## 5. Conclusions

The results of this study indicate that d-serine augments salivary secretion by increasing acetylcholine release in the salivary glands.

## Figures and Tables

**Figure 1 biology-12-01227-f001:**
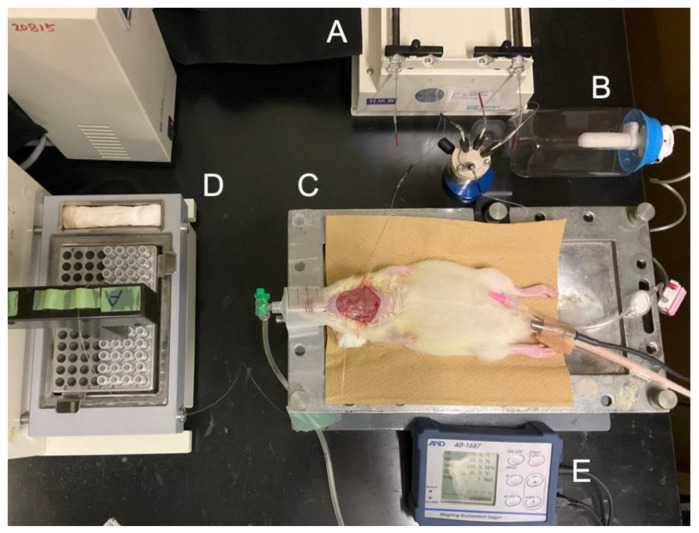
The conditions of a microdialysis experiment. **A**, a micro-syringe pump; **B**, a pressurized pharmaceutical infuser; **C**, a body temperature maintenance device; **D**, a micro-fraction collector; **E**, a rectal thermometer.

**Figure 2 biology-12-01227-f002:**
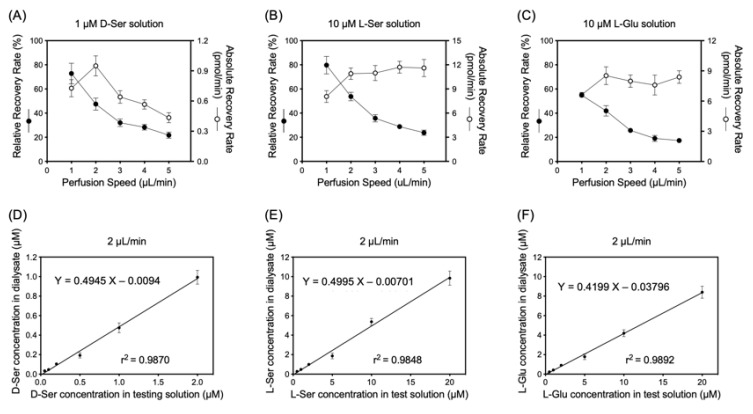
Upper panels (**A**–**C**) indicate relationship between perfusion speed and relative or absolute recovery rate in vitro with 10 µM l-serine (l-Ser), 1 µM d-serine (d-Ser), 10 µM l-glutamate (l-Glu) in testing solution. Relative recovery rate = (concentration in dialysate)/(concentration in testing solution); Absolute recovery rate = (concentration in dialysate) × (perfusion speed). Lower panels (**D**–**F**) show relative recovery rates with different l-Ser, d-Ser, or l-Glu concentrations in testing solution using 2 µL/min perfusion speed. Values represent mean ± SD of 5 dialysates sampled for either (**A**,**B**) or (**C**).

**Figure 3 biology-12-01227-f003:**
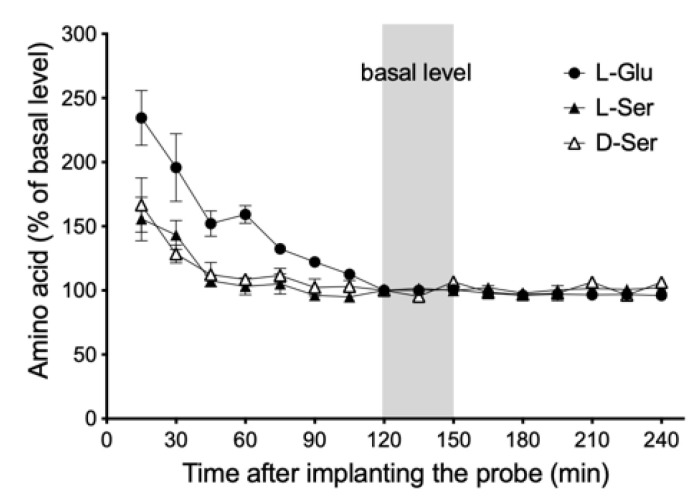
Time course of change in levels of l-serine (l-Ser), d-serine (d-Ser), and l-glutamate (l-Glu) in dialysate after probe implantation. The gray solid line bars show the three fractions of the basal level. Concentration of these amino acids in dialysate maintained an almost steady-state level for 240 min after probe implantation. Values represent mean ± SD in six rats and are expressed as percentages of basal level.

**Figure 4 biology-12-01227-f004:**
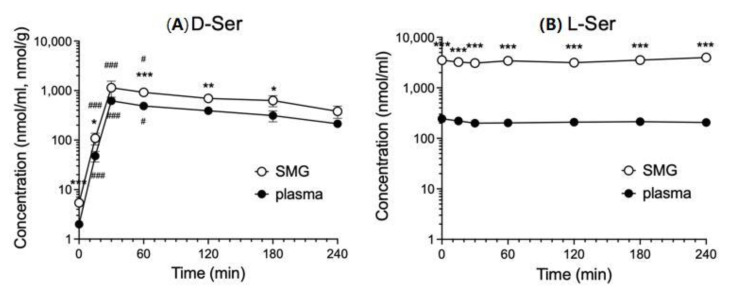
Time course of change in l-serine (l-Ser) or d-serine (d-Ser) levels in plasma and submandibular glands of rats after oral administration (1 mmol/kg). Significantly different from saline (0 min, *n =* 5 each) administered plasma (closed circles, each time *n =* 5) or submandibular gland (SMG) (open circles, each time *n =* 5) according to Dunn’s post-hoc test following Kruskal–Wallis test; # *p* < 0.05 and ### *p* < 0.001. Significantly different from plasma at the same time, according to Dunn’s post-hoc test following two-way repeated measures ANOVA; * *p* < 0.05, ** *p* < 0.01, and *** *p* < 0.001.

**Figure 5 biology-12-01227-f005:**
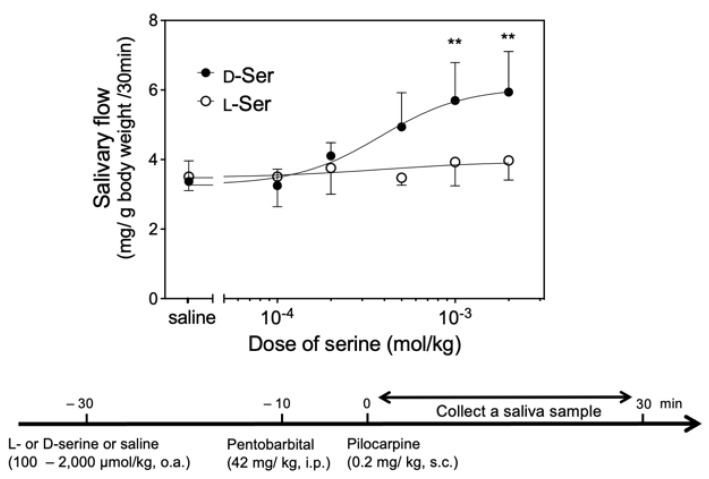
Upper panel indicates salivary secretion by oral administration of d-serine (d-Ser: closed circles, each dose *n =* 5) or l-serine (l-Ser: open circles, each dose *n =* 5) by dose. Significantly different from saline, according to Dunn’s post-hoc test following Kruskal–Wallis test: ** *p* < 0.01. Lower panel shows time schedule of administration of d-Ser, l-Ser, drugs (pentobarbital, pilocarpine), and saline. o.a., oral administration; i.p., intraperitoneal injection; s.c., subcutaneous injection.

**Figure 6 biology-12-01227-f006:**
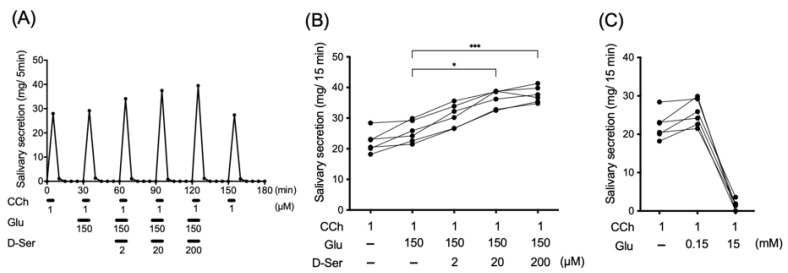
Left panel (**A**) indicates salivary secretion from rat submandibular gland with periodic carbachol (CCh) stimulation using the perfusion system. At 0.5 mL/min perfusion rate, 1 µM CCh was administered over a 5 min period every 30 min. Data represent the mean ± S.D. of 6 experiments. Middle panel (**B**) shows effect of d-serine (d-Ser) on salivary secretion from rat submandibular gland by in-site perfusion system at 30 min after commencement of stimulation by CCh. Significantly different from saline, according to Dunn’s post-hoc test following Friedman test: * *p* < 0.05 and *** *p* < 0.001. Right panel (**C**) shows effect of l-glutamate (Glu) on salivary secretion from rat submandibular gland by in-site perfusion system of l-glutamate by dose at 30 min after commencement of stimulation by CCh.

**Figure 7 biology-12-01227-f007:**
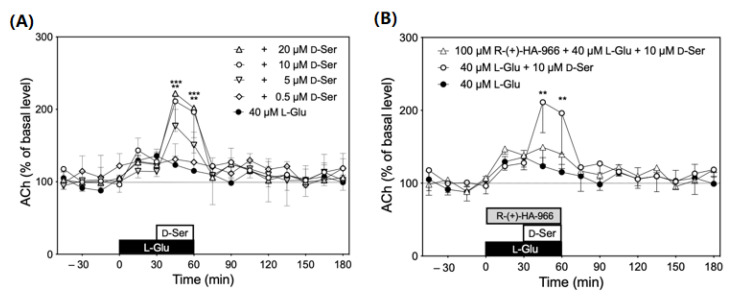
Left panel (**A**) indicates acetylcholine (ACh) release in response to infusion of d-serine (d-Ser, each dose *n =* 5) along with 40 µM l-glutamate (l-Glu). Significantly different from l-Glu alone at same time according to Dunn’s post-hoc test following Friedman test; ** *p* < 0.01 and *** *p* < 0.001. Right panel (**B**) shows effect of R-(+)-HA-966, an antagonist at the glycine site of the NMDA receptor, on ACh release in response to infusion of d-Ser (each dose *n =* 5) along with 40 µM l-Glu. Significantly different from l-Glu alone at same time according to Dunn’s post-hoc test following two-way analysis of variance (ANOVA); ** *p* < 0.01.

**Table 1 biology-12-01227-t001:** The amount of d, l-serine and l-glutamate in plasma, saliva and interstitial fluids in rats.

	d-Ser	l-Ser	l-Glu
plasma	2.53 ± 0.25	217.45 ± 31.25	144.47 ± 10.63
saliva	0.67 ± 0.35	57.86 ± 14.79	20.64 ± 5.02
interstitial fluids	0.21 ± 0.01	5.81 ± 0.33	16.21 ± 3.47

The results are expressed as the mean ± S.D. (nmol/mL, nmol/g) from 5 rats.

## Data Availability

The data that support the findings of this study are available from the corresponding author (M.Y.) upon reasonable request.

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
