# Peer review of "d-Serine Increases Release of Acetylcholine in Rat Submandibular Glands"

_biology, 2023, doi:10.3390/biology12091227_

Round 1
Reviewer 1 Report
The authors have presented a very interesting paper with some clear effects. There are one or two points that could use some clarification though.
1) Why did you use carbachol stimulation (of the acinar cells) rather than electrical stimulation of the nerves? Does the effect not work on the nerves? Presumably what we are seeing here is the dual source of neurotransmitters rather than enhanced release of ACH.?
2) Unless i missed it did you give D-serine infusion alone in section 3.4 fig 5? We probably need this control.
3) presumably your micro perfusion method doesn't greatly alter the composition of the interstitial fluid. What evidence do you have that it doesn't? 2 hours is quite a long time for things to settle down after implantation. Could we see a picture of the setup?
A few minor points:
IN fig 4 could you change p.o to "o.a." for oral adminstration
IN fig 6 the increase in salivary flow with glutamate (t0-t60) looks significantly different to the baseline, is it?
Author Response
Reviewer #1:
The authors have presented a very interesting paper with some clear effects. There are one or two points that could use some clarification though.
Response: We thank the reviewer for his/her careful reading our manuscript and for his/her helpful suggestions.
Specific comments:
- Why did you use carbachol stimulation (of the acinar cells) rather than electrical stimulation of the nerves?
Response: We thank the reviewer for raising this point. It has been confirmed that the effects of electrical stimulation on salivation over a prolonged period of time are difficult to assess accurately due to the large attenuation of the effects. For this reason, carbachol stimulation was used in this study. We apologize because our studies do not show that using acinar suspension cells D-serine led to enhancement of Ca2+ entry in the cells. We would like to carry out this experiment in the future and discuss the points the reviewer raised.
Does the effect not work on the nerves? Presumably what we are seeing here is the dual source of neurotransmitters rather than enhanced release of ACH.?
Response: We thank the reviewer for raising this point. Present study demonstrated that perfusion of D-serine into submandibular glands increased in release acetylcholine level is good agreement with previous studies demonstrated that NMDA receptors on cholinergic nerve terminals stimulates ACh release in brain. We have added sentence as follows; “Furthermore, the present results are in good agreement with previous studies showing that the NMDA receptors located presynaptically and that acetylcholine is released by NMDA receptor stimulation in the brain [35-38]. Further studies on the localization of NMDA receptors in the salivary glands are needed.” Please see line 318-322.
- Unless I missed it did you give D-serine infusion alone in section 3.4 fig 5? We probably need this control.
Response: We thank the reviewer for raising this point. We apologize because our studies do not show that D-serine alone led to saliva secretion. Our studies suggest D-serine dose-dependently enhanced saliva secretion along with glutamate. D-Serine is an obligatory co-agonist for glycine site of NMDA receptor. The coexistence of D-serine and glutamate is essential for the activation of NMDA receptors. In contrast, the concentration of glutamate in blood is saturated under physiological conditions. In fact, we found that exogenously administered D-serine alone increased the salivary secretion in rat submandibular glands (Figure. 5).
- presumably your micro perfusion method doesn't greatly alter the composition of the interstitial fluid. What evidence do you have that it doesn't? 2 hours is quite a long time for things to settle down after implantation. Could we see a picture of the setup?
Response: We thank the reviewer for this suggestion. We, in fact, are aware of the importance of checking the composition of the extracellular fluid after 2 hours perfusion (Figure 3). In microdialysis in the brain, it takes about three hours after probe insertion for the composition of the extracellular fluid to reach a stable state [1]. Immediately after probe insertion, a large amount of amino acids derived from blood and destroyed tissue of salivary glands are considered to be released into the intercellular space. We added a picture of the setup for microdialysis (Figure. 1).
- Bodner, O.; Radzishevsky, I.; Foltyn, V.N.; Touitou, A.; Valenta, A.C.; Rangel, I.F.; Panizzutti, R.; Kennedy, R.T.; Billard, J.M. and Wolosker, H. D-serine signaling and NMDAR-mediated synaptic plasticity are regulated by system A-type of glutamine/ D-serine dual transporters. Journal of Neuroscience 2020, 40, 6489-6502.
A few minor points:
IN fig 4 could you change p.o to "o.a." for oral administration.
Response: We thank the reviewer for raising this point, and correcting us. We have changed it. Please see figure 5.
IN fig 6 the increase in salivary flow with glutamate (t0-t60) looks significantly different to the baseline, is it?
Response: We thank the reviewer for raising this point. In this study, glutamate alone was used as a control for statistical analysis (Dunn’s post-hoc test following ANOVA). Therefore, it was not examined whether glutamate increases acetylcholine release compared to baseline. In accordance with the reviewer's remarks, statistical analysis of whether glutamate increases acetylcholine levels compared to baseline (t=0) showed no significant differences (Friedman test, p=0.66).

Reviewer 2 Report
The study employed animal experiments using male Wistar rats and various sample preparation techniques to analyze amino acids and interstitial fluids in the submandibular gland. However, Salami slicing results in the fragmentation of research, where significant findings are split into multiple publications, is discouraged. Both studies focus on the effects of D-serine on a specific physiological process (neurotransmission in 10.3390/biology11030390 and salivary secretion in the current study) and involve the use of experimental techniques such as perfusion and microdialysis.
I cannot verify that each study likely contributes unique findings and insights to their respective fields, given the overlapping information in the body and methodology.
Good
Author Response
Reviewer #2:
The study employed animal experiments using male Wistar rats and various sample preparation techniques to analyze amino acids and interstitial fluids in the submandibular gland. However, Salami slicing results in the fragmentation of research, where significant findings are split into multiple publications, is discouraged. Both studies focus on the effects of D-serine on a specific physiological process (neurotransmission in 10.3390/biology11030390 and salivary secretion in the current study) and involve the use of experimental techniques such as perfusion and microdialysis.
I cannot verify that each study likely contributes unique findings and insights to their respective fields, given the overlapping information in the body and methodology.
Response:
We do not believe that the reviewer's criticism that the content of the present paper overlaps with one of our earlier paper applies at all. The reviewer's understanding of both our earlier and present study is fundamentally incorrect. We would like the reviewer to carefully read both the earlier paper and the present manuscript, and then to specifically point out for what reasons he/she has determined that this manuscript contains “overlapping information” with our earlier study (10.3390/biology11030390), in other words, plagiarism. We would definitely appreciate a response from the reviewer in order to restore our reputation, which has been damaged by being rated as plagiarism by him/her.
To help the reviewer's understanding, we provide a summary of the earlier study and the present study as follows.
The earlier study (10.3390/biology11030390) showed that in three major salivary glands of rat 1) three D-amino acids (D-serine, D-aspartate, and D-alanine) were detected, 2) all these D-amino acids are ligands for the NMDA receptor, 3) serine racemase, a D-serine synthase, and D-amino acid oxidase, a catabolic enzyme of neutral D-amino acid, and D-aspartate oxidase, a catabolic enzyme of D-aspartate, are expressed at protein and gene levels, and 4) the NMDA receptor subunits (NR1, NR2D) are expressed at the protein and gene levels. The earlier study is the first to comprehensively determine the amounts of D-amino acids, enzymes for their metabolism or catabolism, and NMDA receptors in salivary glands. In the earlier study, we did not examine the effects of any D-amino acids on neurotransmission, much less used experimental techniques such as perfusion or microdialysis.
On the other hand, the present study is the first to focus on salivary secretion by D-serine. In other words, we found 1) that exogenously administered D-serine increased its concentration in rat submandibular glands and then the salivary secretion, 2) that perfusion of D-serine into the submandibular glands through submandibular gland artery increased salivary secretion, and 3) that injection of D-serine through microdialysis probe enhanced acetylcholine release in extracellular spaces in rat submandibular glands.
There is a splitting of results between this submission and a previously published one (doi: 10.3390/biology11030390). It is generally preferable to present the findings of a research project comprehensively and cohesively. Splitting results across multiple publications can lead to difficulties in interpreting, replicating, and synthesis of the research.
Response:
This point by the reviewer is also due to his/her fundamental lack of understanding of both the earlier and the present study. Based on the results obtained from the earlier study showing the comprehensive analysis of D-amino acids, their metabolizing and catabolic enzymes, and their receptors in rat salivary glands, this study focused on D-serine and then provided its “kinetics”, “function”, and “mechanism of action” in the salivary glands. This paper presents “comprehensive and cohesive new findings” obtained from a new viewpoint. We have never published our results across multiple publications.
Again, we would definitely encourage the reviewer to specifically point out which parts of this manuscript are “splitting results across multiple publications”.

Reviewer 3 Report
Dear colleagues!
Your research is well-planned and looks to be useful for science and future clinic.
But please clarify some points
1. What was your null hypothesis?
2. How did you calculate sample size for your research?
3. Line 81. How mane rats were in your research?
4. Line 94. Why did you choose pentobarbital in anesthesia? And how it can be connected with line 115 where you wrote about isoflurane (2%) inhalation?
Am i right that you had few groups? Why did you choose such different drugs and (if) it plays role for gland activity?
5. Please add more in the discussion about Sjögren's syndrome and your results. Can we find any correlations between your research and this syndrome.
Finally, references seems to me are a little bit old, so please add relevant data (may be more modern in 5 years period). It will help you to update the introduction also.
Author Response
Reviewer #3:
Your research is well-planned and looks to be useful for science and future clinic.
Response: We thank the reviewer for his/her kind remark.
But please clarify some points
- What was your null hypothesis?
Response: We thank the reviewer for raising this point. Many statistical analyses in this study require a null hypothesis, which assumes no significant difference between treatments, times, and values. In our experiments, the null hypothesis is that there is no significant difference between plasma levels, salivary gland tissue levels (Figure 4), salivary secretion (Figure 5, 6), and acetylcholine release (Figure 7) before and after D-serine administration.
- How did you calculate sample size for your research?
Response: We thank the reviewer for raising this point. Sample size and power for this study were estimated using PS software with an alpha value of 0.05 and a power value of 0.8 (https://biostat.app.vumc.org/wiki/Main/PowerSampleSize). Based on the results of the preliminary experiment, the delta, which is the minimum difference between the means of the groups to be detected in the main experiment, was determined. PS software finds the number of samples for the t-test, which is a parametric test. We noted that this experiment primarily used nonparametric test.
- Line 81. How mane rats were in your research?
Response: We thank the reviewer for raising this point. We have added the number of rat. Please see line 76.
- Line 94. Why did you choose pentobarbital in anesthesia? And how it can be connected with line 115 where you wrote about isoflurane (2%) inhalation?
Am I right that you had few groups? Why did you choose such different drugs and (if) it plays role for gland activity?
Response: We thank the reviewer for raising this important point. We understand that the same anaesthetic should be applied. In the perfusion experiments, pentobarbital was used as the anaesthetic due to the need to collect saliva. Because microdialysis experiments take a long time, at least 240 minutes from probe insertion to the end of the experiment, we used an inhalation anaesthetic that can maintain a constant anaesthetic concentration. For this reason, different anaesthetic agents were used.
- Please add more in the discussion about Sjögren's syndrome and your results. Can we find any correlations between your research and this syndrome.
Response: We apologize for not being clear about this important matter in the original submission. In this paragraph, we aimed to discuss cells releasing D-serine within the salivary glands. We have added sentence as follows; “The important question arises as to the mechanisms of D-serine release. Previous studies demonstrated that serine racemase and D-serine a located in neurons as well as astrocytes [7, 44, 45]. Under physiological conditions, D-serine acts as a neurotransmitter and is involved in the mediation of synaptic plasticity by NMDA receptors. In contrast, under pathological condition such as injury, D-serine release is switched from neurons to astrocytes [46].” Please see line 333-338.
Finally, references seems to me are a little bit old, so please add relevant data (may be more modern in 5 years period). It will help you to update the introduction also.
Response: We thank the reviewer for raising this point. Mainly in the introduction, where possible we have changed to newer references.

Round 2
Reviewer 3 Report
Dear colleagues!
Thanks for the replies, everything is clear to me now.